# Type IV Collagen Is Essential for Proper Function of Integrin-Mediated Adhesion in *Drosophila* Muscle Fibers

**DOI:** 10.3390/ijms20205124

**Published:** 2019-10-16

**Authors:** András A. Kiss, Nikoletta Somlyai-Popovics, Márton Kiss, Zsolt Boldogkői, Katalin Csiszár, Mátyás Mink

**Affiliations:** 1Institute of Medical Biology, University of Szeged, Somogyi B. u. 4, H-6720 Szeged, Hungary; kiss.andras.attila@med.u-szeged.hu (A.A.K.); somlyai-popovics.nikoletta@med.u-szeged.hu (N.S.-P.); mxk044@shsu.edu (M.K.); boldogkoi.zsolt@med.u-szeged.hu (Z.B.); 2John A. Burns School of Medicine, University of Hawaii, 1960 East West Road, Honolulu, HI 96822, USA; katalin@hawaiii.edu

**Keywords:** Type IV collagen, basement membrane, mutation, myopathy, dystrophy

## Abstract

Congenital muscular dystrophy (CMD), a subgroup of myopathies is a genetically and clinically heterogeneous group of inherited muscle disorders and is characterized by progressive muscle weakness, fiber size variability, fibrosis, clustered necrotic fibers, and central myonuclei present in regenerating muscle. Type IV collagen (*COL4A1*) mutations have recently been identified in patients with intracerebral, vascular, renal, ophthalmologic pathologies and congenital muscular dystrophy, consistent with diagnoses of Walker–Warburg Syndrome or Muscle–Eye–Brain disease. Morphological characteristics of muscular dystrophy have also been demonstrated *Col4a1* mutant mice. Yet, several aspects of the pathomechanism of COL4A1-associated muscle defects remained largely uncharacterized. Based on the results of genetic, histological, molecular, and biochemical analyses in an allelic series of *Drosophila col4a1* mutants, we provide evidence that *col4a1* mutations arise by transitions in glycine triplets, associate with severely compromised muscle fibers within the single-layer striated muscle of the common oviduct, characterized by loss of sarcomere structure, disintegration and streaming of Z-discs, indicating an essential role for the COL4A1 protein. Features of altered cytoskeletal phenotype include actin bundles traversing over sarcomere units, amorphous actin aggregates, atrophy, and aberrant fiber size. The mutant COL4A1-associated defects appear to recapitulate integrin-mediated adhesion phenotypes observed in RNA-inhibitory *Drosophila*. Our results provide insight into the mechanistic details of COL4A1-associated muscle disorders and suggest a role for integrin-collagen interaction in the maintenance of sarcomeres.

## 1. Introduction

Basement membranes (BMs) are 80–100 nm thick, sheet-like extracellular matrices underlying epithelial and endothelial cells in muscular, neural, vascular and adipose tissues. BMs contain major and minor proteins, including type IV collagen, laminin, nidogen/entactin, perlecan, and integrins [1]. Integrity of the BM is a prerequisite for skeletal muscle stability. Research revealed that several muscular dystrophy types may develop as the result of the loss of cell-BM anchorage [2]. Causative gene mutations were reported in the *laminin-A2* (*LAMA2*) gene [3,4]. Mutations in the collagen VI genes were linked to Ullrich CMD, to the milder Bethlem myopathy and to autosomal recessive myosclerosis myopathy [5], while integrin A7 (*ITGA7*) and A5 (*ITGA5*) mutations were shown to be associated to a rare form of CMD [6,7,8].

Mammals harbor three pairs of head-to-head oriented type IV collagen genes, whereas *Drosophila* has one, the *col4a1* and *col4a2* loci in the same genomic organization [9]. Mutations in the *COL4A3*, *A4* and *A5* genes associate with Alport Syndrome, [10,11]. Deletions within the *COL4A5* and *COL4A6* genes are also reported to cause diffuse leiomyomatosis [12]. Heterotrimers with [COL4A1]_2_COL4A2 composition constitute stochiometrically the most abundant mammalian basement membranes. *Col4a1* or *Col4a2* mouse mutants develop complex, systemic phenotypes affecting the central nervous, ocular, renal, pulmonary, vascular, reproductive and muscular systems [13,14,15], as in humans [16]. Severe muscular phenotypes were reported in patients with certain *COL4A1* mutations as part of a multi–system disorder referred to as hereditary angiopathy with nephropathy, aneurysms, and muscle cramps (HANAC) [17,18]. Some patients with *COL4A1* mutations were also diagnosed with Walker–Warburg Syndrome or Muscle–Eye–Brain disease, a distinct form of CMD [19]. *Col4a1^G498V/G498V^* homozygous mice are severely affected by muscular dystrophy including muscle mass decrease, fiber atrophy, centronuclear fibers, fibrosis, focal perivascular inflammation, and intramuscular hemorrhages [20].

In HANAC Syndrome, mutations proved to affect multiple putative integrin binding sites within the COL4A1 protein [21,22]. Proper integrin concentration/function was shown to be required for maintenance of the sarcomere structure [23]. *Drosophila* integrin null mutants loose sarcomeres completely [24], where the ubiquitous integrin dimer is composed of one of the alpha PS subunits combined with the beta PS protein [25]. Conditional RNAi knockdown of *talin*, *alpha-actinin*, *integrin-linked kinase*, *alpha PS2* and *beta PS* integrins revealed a spectrum of phenotypes affecting Z-disc proteins that were dislocated and deposited across the sarcomere, and Z-disc streaming characteristic of myopathic/dystrophic conditions [26].

We have identified an allelic series of conditional, temperature-sensitive *col4a1* mutations in *Drosophila.* The *col4a1^−/−^* homozygotes are embryonic lethal while *col4a1^+/−^* heterozygotes are viable and fertile at permissive temperature of 20 °C, but perish at restrictive condition of 29 °C. In these mutants we have demonstrated severe myopathy [9], irregular and thickened BM, detachment of the gut epithelial and visceral muscle cells from the BM [27], intestinal dysfunction, overexpression of antimicrobial peptides, excess synthesis of hydrogen peroxide and peroxynitrite [28]. In epithelial cells of Malpighian tubules we demonstrated fused mitochondria, membrane peroxidation [29], actin stress fibers and irregular integrin expression [30]. Our results indicated that muscular dystrophy may also be present in *col4a1* mutant *Drosophila* [31].

In order to characterize muscle phenotype in the *col4a1* allelic mutant series we have determined the mutation sites, in immunohistochemistry experiments focused on the striated oviduct muscle we noted aberrant sarcomeres, altered integrin expression and localization, Z-disc disorganization and streaming, fiber size disproportion and atrophy. Results collectively indicate that in mutants dystrophic muscle phenotype appears to originate from compromised integrin interactions with aberrant COL4A1, and supports a role for type IV collagen as part of integrin-mediated muscle cell adhesion.

## 2. Results and Discussion

### 2.1. Characterization of col4a1 Mutation Sites

We analyzed the DNA sequence of PCR products of the *col4a1* gene using genomic DNA isolated from our series of *col4a1^+/−^* heterozygotes. Consistent with the ethyl-methane-sulfonate (EMS) mutagenesis used to generate these mutants [9], by which the product, *O*-6-ethylguanosine, mispairs with T in the next round of replication, causing a G/C to A/T transition, we identified transition in all mutant loci (Appendix A). Resulting from these transitions glycine substitutions by aspartic acid, glutamic acid or serine were identified in the mutants; hereafter we refer to the mutations as displayed in Table 1. In our prior study we reported five mutations in the *col4a1* gene using the *DTS-L3* allele; the G552D and the A1081T amino acid substitutions arose by transitions, similarly to the G to A transition within the 3′ UTR region of the gene. The K1125N, G1198A amino acid substitutions are results of transversions [9]. The A1081T, the 3′ UTR transitions and the K1125N, G1198A transversions occurred in all sequenced mutants. We therefore concluded that these mutations are carried by the balancer chromosome, *CyRoi*. These balancer chromosome mutations do not contribute to the phenotype, given the most robust phenotypical feature of the mutants, the dominant temperature sensitivity is not influenced by exchange of the *CyRoi* chromosome into the chromosome carrying the *col4a2::GFP* transgene [9].

Two pairs of alleles were found to carry the same mutation. The *DTS-L2* and *DTS-L3* lines harbor the same G552D substitution and we refer to these as *col4a1^G552D1^* and *col4a1^G552D2^*. Similarly, the *DTS-L4* and *DTS-L5* lines both carry the G1025E substitution and were designated as *col4a1^G1025E1^* and *col4a1^G1025E2^* alleles. These data confirmed our previous genetic results, for example: The *col4a1^G552D1^* and *col4a1^G552D2^* lines did not complement each other, but complemented the other variants by interallelic complementation [9]. The distribution of the mutant sites at protein level is demonstrated in Figure 1.

The series of mutations proved to cover the collagenous region of the *col4a1* gene that corresponds to amino acid 233 up to 1393 within the 170 kDa COL4A1 protein. The *col4a1^G233E^* allele is within the peptide GFP**G/E**EKGERGD (the G to E substitution in bold), a putative integrin binding site in the COL4A1 protein [21]. The mutation sites in the *col4a1^G552D1^* and *col4a1^G552D2^* lines localize in the immediate proximity of the peptide GLPGEKGLRGD, that resembles the integrin binding site in the COL4A2 protein in the triple helical model made up of [COL4A1]_2_COL4A2 protomers in *Drosophila*, as proposed by us [9] and presented graphically in Figure 1.

### 2.2. Loss of Sarcomere Structure in col4a1 Mutants

The single-layer striated muscle fibers circumventing the common oviduct were analyzed in all mutants by confocal fluorescence microscopy. Normally, these muscle fibers present regular sarcomere structure as demonstrated in Figure 2.

In the mutants, however, the most conspicuous phenotype was the loss of sarcomeres at restrictive temperature, 29 °C (Figure 3, B4 through G4, Appendix A, A4, B4), whereas in wild-type control flies normal sarcomere structure and striation was present at both 20 °C and 29 °C (Figure 2, Figure 3, A1, A4). In mutants at 29 °C, parallel ordered enhanced actin staining intensity areas were present within the muscle fibers that extended over areas larger than a single sarcomere, resembling actin stress fibers or excess actin cross-linking (Figure 3, white rectangles in B4 through G4, Appendix A, A4, B4). Beyond these areas, amorphous, intensive actin staining aggregates appeared in the sarcoplasm (Figure 3, white arrows in B4 through G4, Appendix A, A4, B4). An additional prominent phenotype of the *col4a1* mutants at 29 °C was the irregular and uneven COL4A1 deposition in the individual muscle fibers (Figure 3, white arrowheads, B5 through G5, Appendix A, A5, B5), while in wild-type controls homogenous COL4A1 staining was present at 29 °C (Figure 3, A5). In the isoallelic mutants *col4a1^G552D2^* and *col4a1^G1025E2^* the same COL4A1 staining pattern was observed (Appendix A) as in the other lines of the allelic series (Figure 3). Muscle fibers of the common oviduct in wild-type animals harbor morphologic features of the striated muscles, present the regular register of I and A bands and sarcomeres are bordered by Z-discs (Figure 1, Figure 3, A1–A6). We therefore conclude that the compromised sarcoplasmic morphology and uneven COL4A1 staining/localization is a general phenotype of our series of *col4a1* mutations.

The control flies do not alter their actin accumulation in the muscle fibers at permissive or restrictive temperatures, as measured by quantitative fluorescence confocal microscopy. The presence of *col4a1* mutation seems to exert a non-consequent influence on the actin content of the muscle fibers at both permissive or restrictive conditions. Statistically significant downregulation of actin synthesis, compared to wild-type control flies occurred in *col4a1^G233E^*, *col4a1^G552D1^*, *col4a1^G1025E2^* mutants at 20 °C, whereas mutant *col4a1^G1393E^* synthesized actin at higher concentration (Figure 4). Under restrictive conditions we observed univocal decrease of actin concentration in *col4a1^G552D1^*, *col4a1^G1043S^* and *col4a1^G1393E^* mutants (Figure 4).

The mutant *col4a1^G233E^* accumulates actin at 29 °C, similarly to the *col4a1^G1025E2^* allele, whereas the actin concentration in the isoallelic *col4a1^G1025E1^* mutant remains unchanged (Figure 1, panel C’). In the rest of the mutants we recorded elevated actin concentration (Figure 1, panel C’). Either up- or downregulation of actin expression in common oviducts seems disadvantageous given all mutants are female sterile and do not lay eggs at restrictive temperature [9].

### 2.3. The Myopathic Phenotype Co-Segregates with the Mutation-Carrying Chromosome

The head-to-head pair of *col4a1* and *col4a2* genes localize to the 25C band of the second chromosome in *Drosophila*. If two different dominant temperature-sensitive (DTS) mutations are present in *trans* configuration the compound heterozygotes provide viability by interallelic complementation [9]. In order to determine to what extent compound heterozygotes can recapitulate the dominant temperature-sensitive phenotype affecting sarcoplasmic actin morphology, we have generated the *col4a1^G233E/G1025E1^* double mutant and its reciprocal pair *col4a1^G1025E1/G233E^*. In both compound heterozygotes, the sarcomere structure was lost, actin bundles developed (Figure 2 A,D, white rectangles), intensively staining actin aggregates were deposited (Figure 5 A,D, white arrows), and the COL4A1 protein was detected in uneven and irregular pattern (Figure 5 B,E, white arrowheads). These results indicate that in *col4a1 ^+/−^* heterozygotes, compromised sarcoplasmic actin morphology and aberrant COL4A1 expression and localization are linked to *col4a1* mutations, are independent from the genetic context, and are not a secondary effect of increased temperature.

### 2.4. Z-disc Disintegration, Streaming and Aberrant Integrin Expression in col4a1 Mutants 

In muscles of wild-type *Drosophila*, integrin is expressed at the muscle attachment sites and appears as punctate staining at the costameres aligned with Z-discs [23]. In order to determine the exact position of the Z-discs in the muscle fibers of the oviduct, we used antibodies against the scaffold protein kettin as a morphological marker. In wild-type controls kettin staining appeared as parallel-ordered lines in each fiber perpendicular to the long axis delineating the sarcomeres and proper striation at both permissive and restrictive temperatures (Figure 3 A1,A4). Immunohistochemistry using anti-integrin antibodies provided the same staining pattern in close localization as observed for kettin (Figure 6 A2,A5, and overlays (Figure 6 A3,A6), confirming integrin localization to the Z-discs in muscle fibers of the oviduct.

In *col4a1* mutant, we observed aberrant integrin expression in the epithelial cells of the Malpighian tubules [30], and also surmised irregular integrin deposition in muscle fibers. In mutant oviductal muscle fibers the Z-disc structure, delineated by integrin expression, was disrupted and formed a zig-zag pattern, Z-disc material appeared torn across a large part of sarcomere, and integrin staining was deposited randomly within the sarcomere as dots, consistent with the muscle pathology of Z-disc streaming (Figure 6, white arrows, panels B–D), in which Z-disc structure is disrupted and Z-disc material appears ripped out and deposited within the sarcomere (Figure 6, yellow arrows, panels B–D). These phenotypic features were enhanced by incubating the mutants at restrictive temperature: Areas expressing excess integrin (Figure 6, panels B–D, red arrows), or depositing integrin scarcely (Figure 6, panels B–D, blue arrows). Z-disc streaming, intrasarcomeric integrin deposition and uneven expression seems to be a general feature of *col4a1* mutants, as the same phenotype was observed in the rest of the members of the allelic series (Appendix A).

We have recently reported a further phenotypic manifestation of the mutants, the ectopic assembly and transition of Z-discs the anisotropic (A) band at the level of M-discs at statistical significance in the mutants [32]. These observations collectively suggest the requirement of proper COL4A1 protein in regular integrin expression and explain female sterility of the mutants that do not lay eggs at 29 °C [9].

### 2.5. Fiber Atrophy and Fiber Size Diversity

We measured the diameter of the individual muscle fibers and the most frequent value was found to correspond to 8 µm both in mutant *col4a1^G233E^* line and wild-type (*Oregon*) controls, incubated at permissive temperature (Figure 7). Incubation of mutant animals at 29 °C shifted the diameters of the muscle fibers toward smaller values. We observed the same phenomenon in the rest of the lines of the allelic series (Appendix A). The ratio of the muscle fibers with diameters below 8 µm up to 4 µm increased two-threefold, by 12%–34% in the mutants, whereas the same ratio in control flies remained 30% at both temperatures with the majority of fiber diameters in the range of 7–8 µm and only 4%–6% of 6 µm as the smallest value (Figure 7, Appendix A, Table 2). These data showed size heterogeneity, wasting and atrophy of muscle fibers, characteristic features of dystrophic muscle in the *col4a1* mutant lines.

## 3. Conclusions

In our mutant series, glycine substitutions by large, charged or polar amino acids, glutamate, aspartate and serine, occurred within the *col4a1* gene by transition of the second guanine nucleotide to adenine consistent with the mutagen EMS. Two isoallelic variants *col4a1^G552D1^, col4a1^G552D2^,* and *col4a1^G1025E1^, col4a1^G1025E2^* alleles were identified. The importance of the Gly552 residue is reflected by the fact that a recent EMS mutagenesis resulted in the isolation of the same, temperature-sensitive, dominant-negative *col4a1^G552D^* allele [33]. Genotype–phenotype relationships explored in over hundred *COL4A1* mutants identified in patients and in murine models revealed that the position of the mutation and not the biochemical properties of the substituting amino acid seems to have a greater impact on the phenotype and disease severity [34]. In our *Drosophila* mutant series, however, regardless of the position of the mutation within the collagenous domain of the *col4a1* gene, we observed similar phenotypic defects.

As the oviduct, and also the larval body wall muscle phenotype [9], compromised actin organization and deposition, loss of sarcomere structure were noted in all alleles studied, features that are common in myopathic or dystrophic conditions, with disintegrating muscle sarcomeres together with disintegration and streaming of Z-discs [35]. These morphologic changes impact the function of the common oviduct as females become sterile and do not lay eggs [9]. Conditional knockdown of genes in *Drosophila*, involved genes in integrin mediated adhesion, including *talin*, *alpha-actinin*, *integrin-linked kinase*, *alpha PS2* and *beta PS* integrins result in the common phenotype of Z-disc streaming [26], similar to our *col4a1* mutant series, indicating functional interdependence. Similarly, Walker–Warburg Syndrome is diagnosed as a monogenic trait in patients carrying mutations in several genes [19,36]. Importantly, in *Drosophila* mutants defective in protein O-mannosyltransferases the symptoms of the Walker–Warburg Syndrome were identified; as part of the mutant phenotypes Z-disc streaming, actin filament disorganization and bundle formation were reported also [36].

The human myopathic/dystrophyc conditions marked by Z-disc streaming and sarcomeric disorganization involves genes that encode components of integrin-mediated adhesion markedly by genetic reasons. The *Ilk^−/−^* mouse embryos die during peri-implantation stage due to impaired epiblast polarization and F-actin accumulation at integrin attachment sites [37]. Knockout mutants for the integrin beta subunits and for majority of the alpha subunits have been constructed with phenotypes ranging from a complete block in preimplantation, through developmental defects to perinatal lethality, demonstrating the specificity of each integrin. Muscular dystrophy was observed in patients with *ITGA5* or *ITGA7* mutations [8]. Z-disc streaming, however, was not reported in association of *ITGA5* or *ITGA7* mutations. Mouse mutants of *talin 1* or *talin 2* perform myopathy and disassembly of the sarcomeres [38]. However, the embryonic lethal mutation in the *Drosophila rhea* gene encoding talin recapitulate the phenotype of the integrin beta PS mutations, demonstrating their functional similarities [25]. These results indicate that genes involved in integrin-mediated adhesion are essential, their homozygous recessive or null mutations are often lethal. The conditional lethality of the temperature-sensitive, heterozygous *col4a1 Drosophila* mutant series allowed manifestation of phenotypic elements that would be non-explorable in humans or mice, such as the disrupted sarcomeric cytoarchitecture and Z-disc streaming that support a role for COL4A1 in integrin mediated adhesion. In conclusion, our *Drosophila* mutant series may serve as an effective model to uncover the mechanisms by which *COL4A1* mutations result in disrupted myofiber-basement membrane interactions and compromised muscle function and provide biomarkers to explore during therapeutic approaches.

## 4. Materials and Methods

### 4.1. PCR Amplification and Sequencing

The algorithm of Primerfox was used to design sequence specific primers for the *col4a1* gene (Table 3). The amplification reaction was carried out with the aid of KAPA Taq polymerase and Fermentas dNTP mix (Thermo Scientific, Vilnius, Lithuania) guided by a touchdown PCR protocol. The initial denaturation at 94 °C lasted for 150 s, followed by 30 cycles of 93 °C for 15 s, then 65 °C (−0.6 °C/cycle) for 15 s and 72 °C for 45 s. The final elongation step at 72 °C was allowed to run for 180 sec. The lengths of the products were checked on 1% agarose gel, followed by cleanup on silica columns (ZenonBio, Szeged, Hungary). The DNA samples containing the appropriate primers were sent to Eurofins Genomics for sequencing. The same PCR fragments originating from different reactions were read multiple times in both directions to ensure reliability of the results. The received sequence information was aligned to the database of NCBI with the Blast algorithm.

### 4.2. Maintenance of Drosophila Strains

Wild-type *Oregon* flies and *col4a1* mutant stocks were maintained at 20 °C and 29 °C on yeast–cornmeal–sucrose–agar food, completed with the antifungal nipagin. The mutant stocks were kept heterozygous over the *CyRoi* balancer chromosome that prevents recombination with the mutation-carrying homolog, thus the genotype of the mutants presented here is: *col4a1^−^ +/col4a1^+^ CyRoi.*

Common oviducts were removed under carbon dioxide anesthesia from adults that were grown at both permissive and restrictive temperature for 14 days. Dissected common oviducts were fixed in 4% paraformaldehyde dissolved in phosphate buffered saline (PBS) for 10 min, washed tree times in PBS, permeabilized for 5 min in 0.1% (w/v) Triton X dissolved in PBS and washed tree times in PBS. Blocking was achieved for in 5% BSA dissolved in PBS for 1 h, and washed tree times in PBS. Trans-heterozygous strains were generated by crossing two *col4a1^+/−^* heterozygotes selecting for the loss of the balancer chromosome *CyRoi.*

### 4.3. Immunostaining and Antibodies

Nuclei in the dissected common oviducts were counter-stained by 1 µg/mL 4′,6-diamino-2-phenylindol (DAPI) in 20 µL PBS, 12 min in dark. F-actin was stained by 1 unit Texas Red^TM^-X Phalloidin (ThermoFisher) in 20 µL PBS for 20 min. Integrin dimer staining was achieved by an equimolar mixture consisting of both anti-integrin monoclonal antibodies (mouse, Developmental Studies Hybridoma Bank) that recognize alpha PS I or alpha PS II subunits. Mouse antibody against *Drosophila* COL4A1 protein was generated by Creative Ltd, Szeged, Hungary. Primary mouse antibodies were visualized by 1 µL F(ab’) 2-Goat Anti-Mouse IgG (H+L) Cross Adsorbed Secondary Antibody conjugated with Alexa Fluor 488 (ThermoFisher) in 20 µL PBS for 1 h or 1 µL Goat Anti-Mouse IgG (H+L) Cross Adsorbed Secondary Antibody, Alexa Fluor 350, in 20 µL PBS for 1 h.

### 4.4. Confocal Microscopy

Photomicrographs of the common oviducts were generated by confocal laser scanning fluorescence microscopy (Olympus Life Science Europa GmbH, Hamburg, Germany). Microscope configurations were set up as described [29]. Briefly, objective lens: UPLSAPO 60x (water, NA: 0.90); sampling speed: 8 µs/pixel; line averaging: 2x; scanning mode: sequential unidirectional; excitation: 405 nm (DAPI), 543 nm (Texas Red) and 488 nm (Alexa Fluor 488); laser transmissivity: 7% were used for DAPI, 15% for Alexa Fluor 488 and 20% for Texas Red.

Quantitative evaluation of the fluorescence light intensities of the 543 nm Texas Red signals was achieved by the FLUOVIEW FV1000 program of Olympus, system version 4.2.1.20. We generated ~20 independent images for each wild-type control and mutant samples thus analyzing over 400 photomicrographs. Statistical analysis was carried out by analyzing raw confocal fluorescent light intensities. Signal amplification was applied in images for morphologic analysis.

### 4.5. Size Determination of the Muscle Fibers

Confocal photomicrographs displaying oviducts were stained by Texas Red™-X Phalloidin and anti-COL4A1 antibody, taken from all mutants and wild-type controls at 20 and 29 °C. Diameters of hundred randomly chosen muscle fibers were measured generating altogether 9000 values. Diameters were calculated as distances between the lateral sides of the individual muscle fibers, perpendicular to the long axis, in areas with unchanged diameters. Bins of diameter intervals differing by one µm were displayed in histograms showing the numbers of the corresponding diameters.

## Figures and Tables

**Figure 1 ijms-20-05124-f001:**
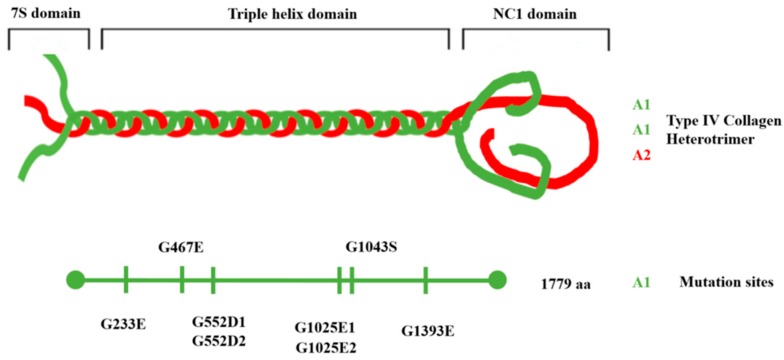
Triple-helical model of type IV collagen trimer with [COL4A1]2COL4A1 composition and distribution of the mutation sites on the COL4A1 protein.

**Figure 2 ijms-20-05124-f002:**
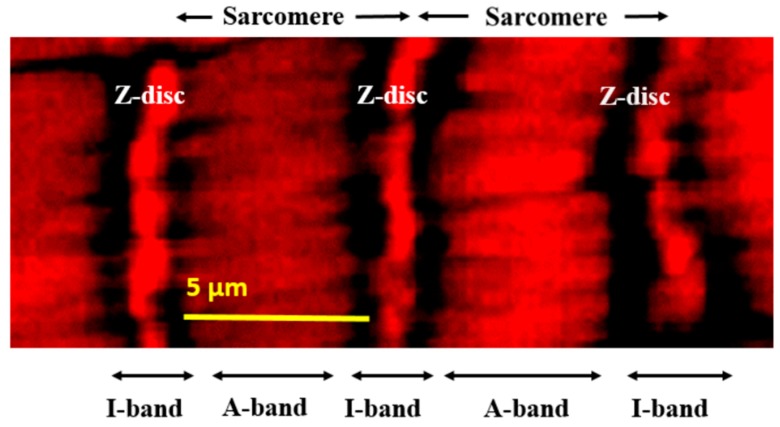
Regular sarcomere units within an oviductal muscle fiber demonstrated at high resolution by fluorescence confocal microscopy and actin staining by phalloidin.

**Figure 3 ijms-20-05124-f003:**
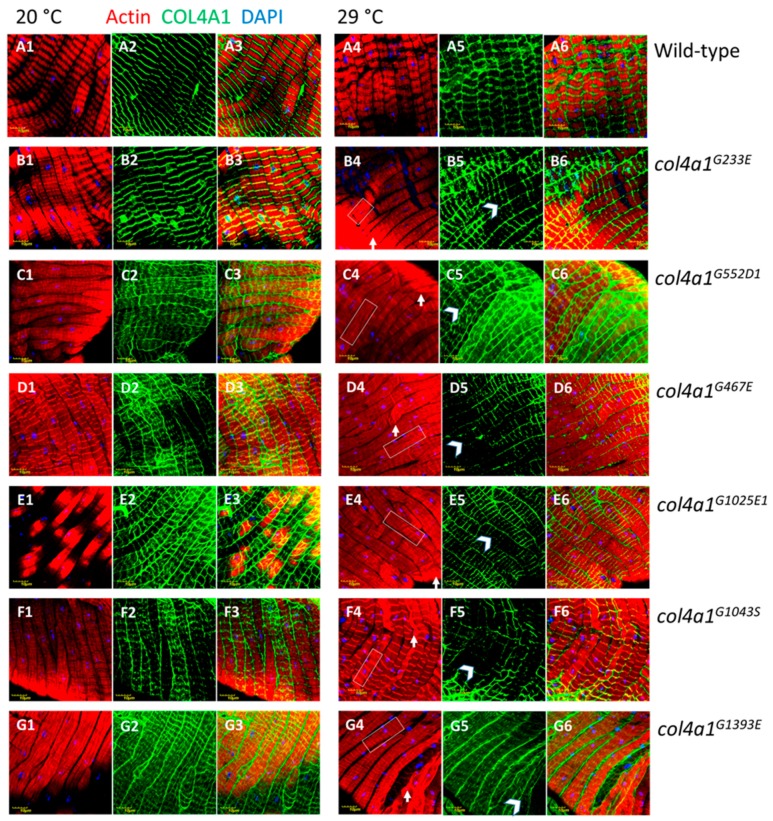
Loss of sarcomeres in *col4a1^G233E^*, *col4a1^G467E^*, *col4a1^G552D1^*, *col4a1^G1025E1^*, *col4a1^G1043S^* and *col4a1^G1393E^* mutant lines at 29 °C (B4 through G4) in comparison with wild-type control (A4). Representative actin bundles (white rectangles in B4 through G4), actin aggregates (white arrows in B4 through G4), uneven COL4A1 expression (white arrowheads in B5 through G5). A3 through G3 and A6 through G6: Overlays of actin and COL4A1 staining. Bars, lower left, 10 micrometers.

**Figure 4 ijms-20-05124-f004:**
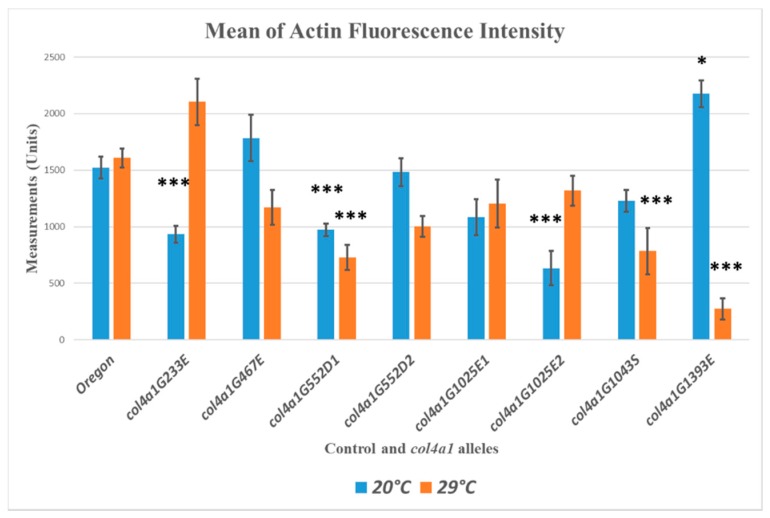
Quantitative measurements of actin content in muscle fibers by fluorescence light intensities of phalloidin labeled muscle. Significance is labelled by asterisk * *p* < 0.05; *** *p* < 0.0005.

**Figure 5 ijms-20-05124-f005:**
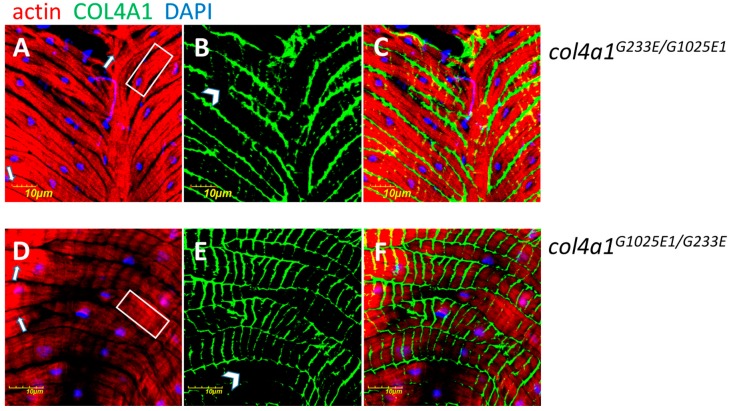
Sarcomeric loss (A, D), actin bundles (A, D, white rectangles), intensively staining actin aggregates (A, D, white arrows), COL4A1 protein in uneven and irregularly deposited fashion (B, E, white arrowheads) in compound heterozygote *col4a1^G233E/G1025E1^* (**A**–**C**) and in reciprocal *col4a1^G1025E1/G233E^* (**D**–**F**) double mutants. C, F: Overlays of actin and COL4A1 stainings. The experiment was performed at 20 °C. Wild-type control provided in Figure 3, upper row A. Bars, lower left, 10 micrometers.

**Figure 6 ijms-20-05124-f006:**
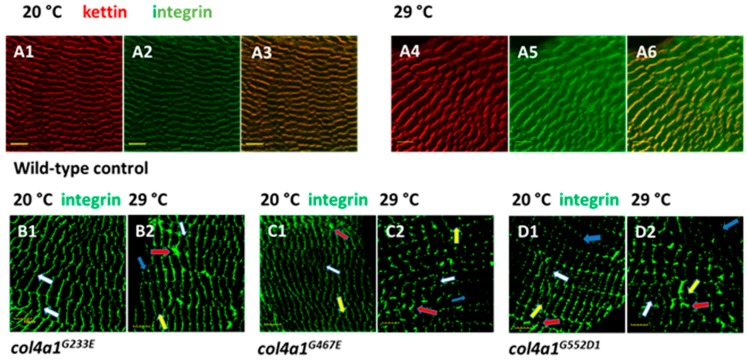
Staining of muscle fibers of common oviduct by the Z-disc marker kettin and integrin in wild-type control (**A1**–**A6**) at permissive and restrictive temperatures. Note unchanged integrin expression at 29 °C (A5) and close vicinity of kettin and integrin depositions revealed by the complementary color orange in overlays (A3, A6). Panels **B**, **C**, **D**: *col4a1^G233E^*, *col4a1^G467E^*, *col4a1^G552D1^* mutants at permissive and restrictive temperatures. Streaming of the Z-discs (white arrows), integrin expression within the sarcomeres (yellow arrows), excess integrin expression (red arrows) or deficient integrin deposition (blue arrows) are noted in the mutants (panels **B**–**D**). Bars, lower left, 10 micrometers.

**Figure 7 ijms-20-05124-f007:**
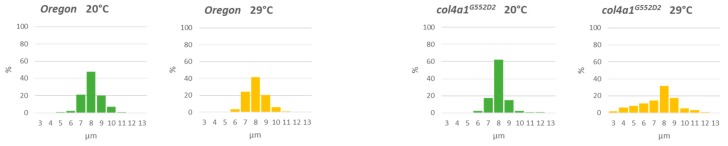
Muscle fiber atrophy measured by the reduced diameter of the fibers in the mutants at 29 °C.

**Table 1 ijms-20-05124-t001:** The mutation sites, former and present designation of the mutant loci.

Former Designation	*a-30*	*b-9*	*DTS-L2*	*DTS-L3*
Present designation Mutation site	*col4a1^G233E^* p.G233E GGA->GAA	*col4a1^G467E^* p.G467E GGA->GAA	*col4a1^G552D1^* p.G552D1 GGC->GAC	*col4a1^G552D2^* p.G225D2 GGC->GAC
Former designation	*DTS-L4*	*DTS-L5*	*DTS-L10*	*b-17*
Present designation Mutation site	*col4a1^G1025E1^* p.G1205E1 GGA->GAA	*col4a1^G1025E2^* p.G1025E2 GGA->GAA	*col4a1^G1043S^* p.G1043S GGG->GAG	*col4a1^G1393E^* p.G1393E GGA->GAA

**Table 2 ijms-20-05124-t002:** Summary of the phenotypes of the *col4a1* mutants.

Allele	*col4a1^G233E^*	*col4a1^G467E^*	*col4a1^G552D1^*	*col4a1^G552D2^*	*col4a1^G1025E1^*	*col4a1^G1025E2^*	*col4a1^G1043S^*	*col4a1^G1393E^*	*OreR, wt*
Sarcomere loss	+	+	+	+	+	+	+	+	-
Z-disc streaming	+	+	+	+	+	+	+	+	-
Irregular integrin expression	+	+	+	+	+	+	+	+	-
Actin bundles	+	+	+	+	+	+	+	+	-
Amorphic actin deposition	+	+	+	+	+	+	+	+	-
Atrophy, fibers with D<8 micrometer at 20 °C	10%	10%	14%	14%	14%	18%	16%	8%	30%
Atrophy, fibers with D<8 micrometer at 29 °C	28%	22%	48%	36%	26%	42%	36%	20%	30%
Fiber size disproportion	+	+	+	+	+	+	+	+	-
Uneven COL4A1 expression	+	+	+	+	+	+	+	+	-

**Table 3 ijms-20-05124-t003:** List of PCR primers used for sequencing the *col4a1* gene.

Name	Sequence
F1a	CACGGATAGTGTACATGAGC
F2a	GCCTTTAGCAAACTCTCTTG
F3	TCCTCGTTTCCCGTCAAACC
F4	TTCAAGGGCAATGCTGGTGC
F5	GGTCTCAATGGTCTGCAAGG
F6	TCCCGGAATGGATGGTTTGC
F7	GAAGGGTGAACCAGGAATGC
F8	TCTGTTGGATACTGCGTAGC
R1	CATAGCTCTCTTCGATTGGC
R2	CTCCCTTCTGTCCCATATCG
R3	CCTTGATACCCATGTCTCCC
R4	AAACCAATGGGTCCGGTTGG
R5	TCACCAGGATAGCCAACAGC
R6	TCTCCCTTAGGTCCATTGCG
R7	CGGCAGTGTGCTATTATAGG
R8	GCATTGTTTCGCATTTAATCGG

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
