# Peer review of "Type IV Collagen Is Essential for Proper Function of Integrin-Mediated Adhesion in Drosophila Muscle Fibers"

_ijms, 2019, doi:10.3390/ijms20205124_

Round 1

Reviewer 1 Report

In this paper Kiss et al systemically characterized the phenotype of single-layer striated muscle of the common oviduct in allelic series of Drosophila col4a1 mutants, arisen by transitions in glycine triplets. They found that those mutations led to loss of sarcomere structure and disintegration and streaming of Z-discs, associated to altered actin bundles and muscle atrophy. The observed defects appear to resemble those observed in integrin knock down models, suggesting a critical role of integrin-collagen interaction in the maintenance of sarcomere integrity.

The experiments presented are convincing, and, although very much descriptive at this stage, the model used might be very useful to study mechanisms by which COL4A1 mutations result in compromised muscle structure and function in human myopathic/dystrophic conditions related to collagen defects, providing further insight to explore therapeutic approaches. Indeed, the conditional lethality of the temperature-sensitive, mutant series used might allow the appearance of phenotypic elements that would be non-explorable in humans or mice.

No major criticisms

Reviewer 2 Report

General comment:

The manuscript describes the role of collagen IV in proper function of integrin-mediated adhesion in Drosophila muscle fibers.

The experimental approach and obtained results are interesting and they are continuation of earlier investigations out by authors and published elsewhere.

The integrin-collagen interactions and role in tissue architecture have been widely studied and discussed in literature and it appears that collagen-binding integrins only have a limited role in adult connective tissue homeostasis (see https://jcs.biologists.org/content/129/4/653). That is mainly due to the limited availability of cell-binding sites in the mature fibrillar collagen matrices.

Noteworthy, Integrin-mediated Cell Adhesion to Type I Collagen Fibrils has been reported previously (http://www.jbc.org/content/279/30/31956).

Nevertheless, the mutagenesis method used in presented studies to identify mutations in within collagen IV doesn’t rule out the possibility that other genes are targeted. Therefore, although the further analyses are interesting, the more advanced and widely available strategies targeting unique loci should be considered to verify the reported findings. Additionally, the quality of the presented figures remains questionable.

Specific comments:

Figure 4. The quantitative measurements of actin content in muscle fibers show over 50percent increase at 20C and, interestingly, approx. 5-fold reduction at 29C for mutant col4a1_G1393E. However, the actin levels seem relatively high at 29C for this mutant when looking at the comparative analysis of different mutants shown at Figure 3.(comparison between A4 and G4). Does this discrepancy result from not reliable measurement of actin content or picture intensity manipulation on Figure 3?

It is not clear why the double mutants exhibit very different phenotypes... (Figure 5). Any explanation/suggestions? Is there any particular reason why only these pairs were tested?

Figure 6. Staining have significant background. In particular, wild-type control at 29C is not clean. In fact, one may argue that the excessive integrin expression (red arrows) or deficient integrin deposition (blue arrows) marked in mutants are also detectable in wild-type control if the image would be enhanced as in case of mutants…

Additionally, although authors claim that integrin expression is not changing at 29C, unfortunately, the picture is not convincing. That may be due to the poor quality of the image or the occurrence of changes on the expression level and localization pattern to some extent. Moreover, the kettin is shown only for wild-type. Why there are no analogous images for mutants showing the localization of kettin? These would seem justified especially since according to authors kettin and integrin colocalize. Would be interesting to determine how the localization of this large modular protein in the Z-disc of insect muscles changes upon type IV collagen mutation. It would show the extent of muscle disorder and provide an evidence for muscles disruption.

The bars indicating size are barely visible for mutants.

Overall, the concept and design is interesting but the experimental design and execution can be questionable. The provided evidence is limited mostly to confocal image analyses that are of poor quality. The images showing staining have background and would be wise to repeat them and provided pictures of better quality.

Round 2

Reviewer 2 Report

The manuscript has been significantly improved. The comments were properly addressed and manuscript is acceptable in the current form.

The scientific soundness although average, due to the significance of the overall context makes the paper more valuable.